# Toward Modular and Flexible Open RAN Implementations in 6G Networks: Traffic Steering Use Case and O-RAN xApps

**DOI:** 10.3390/s21248173

**Published:** 2021-12-07

**Authors:** Marcin Dryjański, Łukasz Kułacz, Adrian Kliks

**Affiliations:** 1Rimedo Labs, 61-131 Poznań, Poland; marcin.dryjanski@rimedolabs.com (M.D.); or lukasz.kulacz@rimedolabs.com (Ł.K.); 2Institute of Radiocommunications, Poznan University of Technology, 61-131 Poznań, Poland

**Keywords:** open RAN, xAPP development, traffic steering, 5G/6G

## Abstract

The development of cellular wireless systems has entered the phase when 5G networks are being deployed and the foundations of 6G solutions are being identified. However, in parallel to this, another technological breakthrough is observed, as the concept of open radio access networks is coming into play. Together with advancing network virtualization and programmability, this may reshape the way the functionalities and services related to radio access are designed, leading to modular and flexible implementations. This paper overviews the idea of open radio access networks and presents ongoing O-RAN Alliance standardization activities in this context. The whole analysis is supported by a study of the traffic steering use case implemented in a modular way, following the open networking approach.

## 1. Introduction

The current world and national economic development will be significantly driven by the practical and wide-scale deployments of 5G cellular networks. Various use cases have been identified and extensively investigated over the last decade, where 5G solutions should incentivize investors in various vertical industry sectors to strengthen their involvement. At the same time, the scientific community all over the world discusses the requirements and challenges for the next technological leap in the wireless communication domain, i.e., the sixth generation of cellular networks [1,2]. One of the key aspects in this context is the increasing role of artificial intelligence tools which are considered for 5G and also for future 6G networks [3,4].

In parallel to this development process, another significant transition is happening in the wireless communication domain that is not part of the main 5G ecosystem [5]. As this will affect the functioning of cellular networks from the mobile network operators’ (MNOs) and infrastructure vendors’ perspective, it will have a very limited influence on the end user. Namely, the architecture of the Radio Access Network (RAN) is rapidly evolving from a solid, black-box approach (also known as silo) towards guaranteeing a high level of openness. In the former case, the hardware manufacturers are typically delivering harmonized and integrated solutions to MNOs, leaving them rather limited possibilities of influencing the internals beyond a typical configuration. In contrast, the so-called Open RAN approach benefits from RAN virtualization and its structural openness [5,6]. Thus, the underlying hardware can be abstracted, allowing for easy and flexible modification of the software installed on it. Please note that this process is one of the consequences of overall network virtualization, of moving various functionalities to the cloud or the edge [7]. Next, the openness of intra-RAN interfaces creates opportunities for flexible software delivery by various telecom vendors. Moreover, the software running on open RAN-supported hardware can be structured in a specific way, where selected algorithms (needed for operating the wireless networks) will be treated as separate applications, managed by a dedicated controller. Following the O-RAN Alliance (here referred to as O-RAN in contrast to Open RAN) specifications, such applications are called xApps or rApps, depending on their time scale of operations [8]. An intelligent controller is often considered a sophisticated entity, equipped with artificial intelligence and machine learning (AI/ML) capabilities. Thus, the MNOs, or in a wider scope service providers (SP) may modify, update, replace or extend selected functions within the RAN protocol stack whenever it is needed.

Such a programmable and highly modular structure of the RAN architecture will have a significant impact not only on the structure of the future wireless networking market, but also on the prospective research and development process in this domain. The presence of AI/ML-equipped modules, as well as independence from the underlying hardware and the necessity of xApp modularity and flexibility, entails the need to redefine the ways new algorithms are invented [9,10]. Let us just mention that it could be necessary to consider the presence of specific kinds of application interfaces (and in consequence, access to specific types of data) between the controller and other applications. Furthermore, a high level of RAN virtualization in the Open RAN concept opens new directions of RAN implementation, which may now be moved to the cloud. This option may be of interest to all MNOs and SPs in the context of deployment of private mobile networks, due to the reduction in the time-to-market and installation costs. Finally, Open RAN is brought forward as one of the important elements in the mobile systems evolution towards the 6G era [11,12]. Seeing the practical realization of the Open RAN concept as a driver in the wireless communication domain, in this paper, we overview the recent achievements in this context. We start with the presentation of RAN transformation from the classic, highly centralized manner to the Open RAN idea. Next, we present the current, yet selected, standardization activities in the O-RAN area, and then we concentrate on the xApp development process. In that spirit, the key contribution of this paper is the description of the traffic steering xApp, which has been implemented in a hierarchical and modular manner [13]. The presented results highlight the benefits of such a modular approach by showing how the AI/ML tools may be used for intelligent management of the xApp functioning, and in consequence, for improving system performance.

## 2. Open RAN Concept

The transformation from the legacy RAN towards the Open RAN can be described in phases. In the first generations of cellular networks, the radio access part was realized in the form of black-box hardware delivered by a single selected vendor. With the evolution of the network architecture, cloud-based approaches have increased their importance. Mainly, the so-called Cloud RAN appeared to be an important element of the 4G network architecture. Evolving further towards Open RAN (and more specifically, O-RAN), the base station (BS) has been split into the centralized unit (CU), distributed unit (DU), and remote unit (RU). Those can be developed by different vendors due to the open interfaces between them, including F1, E1, and Open Fronthaul (see Figure 1). In addition to that, the RAN Intelligent Controller (RIC), is separated from the processing units and allows us to gather radio resource management (RRM) and self-organizing networks (SON) functions, which control the radio resources and network. In the O-RAN concept, this is where the intelligence sits, employing AI models for radio network automation.

### 2.1. O-RAN

In O-RAN, all BS parts mentioned above get the “O-” prefix, meaning that they are adapted to the O-RAN Alliance definition and architecture (see Figure 1). Those entities are connected to RIC via the E2-interface and are called “E2 Nodes”. Furthermore, RIC is split into “near-Real Time RIC” (near-RT RIC) and “Non-Real Time RIC” (Non-RT RIC). The latter one sits at the service and management orchestration system (SMO). This split allows a hierarchization of RRM, differentiating algorithm operations on the timescale of operation [8].

The key characteristics of the O-RAN concept are:disaggregation of BS into granular functions (O-CU/O-DU/O-RU), decoupling of software from hardware, and opening up internal BS interfaces;the above encourages the development of an open and enlarged telco ecosystem with different vendors, such as O-CU vendors, RIC vendors, algorithm/xApp developers, and system integrators;intelligent and holistic management enabled by RIC, where the system intelligence is embedded within the O-RAN architecture using AI models and RRM functions, like Traffic Steering (TS), Interference Management (IM), QoS Management, Load Balancing, etc.

### 2.2. RAN Intelligent Controller (RIC)

RIC is one of the key elements in the O-RAN architecture. It is a platform for which, e.g., xApp software vendors can provide RRM/SON algorithms to allow the optimization of radio resource usage for specific applications. The motivation for RIC is to provide controllability to RAN in order to optimize and improve system performance, taking into account the current state of mobile systems. The complexity of those is increased due to network densification, new spectrum bands, multi-radio-access technology (RAT), and Heterogeneous Network (HetNet) scenarios brought on by 5G. Therefore, the task to optimally allocate radio resources, manage handovers or interference, balance the load between multiple cells and carriers is not trivial [13,14]. With the use of RIC, RRM is decoupled from the RAN stack and enables the implementation of various algorithms collectively utilizing a common set of data. This approach provides holistic control by RRM across all technologies, spectrum bands, cells, and antenna ports. An additional benefit of using this concept is the per-use-case-based management of RAN. According to it, applying policy-based control allows for performance-based decisions adjusted to specific applications.

Figure 2 shows the details of the RIC split into near-RT RIC and non-RT RIC. The former is responsible for handling the near-RT RRM/SON functions (on a timescale between >10 ms and <1 s), such as Mobility Management, IM, etc. The latter handles the high-level/orchestration functions and provides policies to near-RT RIC over the A1 interface. Note that the real-time RRM is still there, embedded in O-DU (e.g., MAC scheduler). The scheduler can be controlled by near-RT RIC only via general policies. To summarize, there are three control loops, RT (<10 ms) handled by O-DU, near RT (>10 ms, <1 s) handled by near-RT RIC, and Non-RT (>1 s) handled by non-RT RIC. It is a hierarchical design, which allows for the separation of resource handling concerns [11]. Near-RT RIC, being responsible for RAN control and optimization, incorporates RRM/SON algorithms (through xApps) and bases its operation on UE- and cell-specific metrics.

As per [15], and as shown in Figure 2, near-RT RIC:provides a database storing the configurations related to E2 nodes, cells, bearers, flows, UEs, and mappings between them;provides ML tools that support data pipelining;provides the messaging infrastructure to support information exchange between xApps and the near-RT RIC framework;provides logging, tracing, and metrics collection from the near-RT RIC framework and xApps to SMO;provides security functions for xApps;supports the conflict resolution function to resolve the potential overlaps that may be caused by various xApps;supports A1 interface used to provide policies and ML model management to near-RT RIC and obtain feedback from its operation using those policies and models.supports E2 interface used to send control/policy messages down to E2 Nodes for resource allocation/prioritization and obtain fine-grained UE/RAN statistics from E2 Nodes.

## 3. RIC and xApps—Towards a CI/CD Approach in RAN Design

Progressive RAN programmability and openness of its architecture, including communication interfaces, move the RAN implementation close to the currently popular trends in efficient code delivery. In particular, the well-known and widely applied concept of continuous integration, continuous delivery (CI/CD) has gained great attention in advanced software development in recent years. This approach assumes that instead of providing new updates of the product in the form of new big releases (or significantly large packages), the final product is upgraded continuously through small patches, while keeping it operational all the time. In turn, the whole programming process has to be specifically designed towards high automation of testing and reliability of code integration, as well as towards high modularity. In addition to virtualization, where the underlying hardware is abstracted, software containerization has recently become highly popular, complementary to the creation of virtual machines. Thus, up-to-date software providers may operate taking profits from both hardware abstraction, and the ability to create isolated spaces within the same operating system for advanced automation and short-time operations. In view of this, the practical realization of the RAN environment may become more programmable, virtualized, and “containerized”.

With this in mind, the Open RAN providers may in turn benefit from modular updating of selected RAN functions, and in general, from continuous and systematic code improvements. The up-to-date solutions in computer science, jointly with the openness and modularity of Open RAN architecture and interfaces, create the basis for a new RAN development paradigm. In particular, it is expected that instead of being part of the black box, the particular functionalities of RAN may be modularized, separated, and may be delivered and upgraded by various software providers. Such functions include, e.g., TS, admission control (see example in [16]), IM, and user paring into layers in massive MIMO networks. As the RRM algorithms operate—to some extent—as separate entities that communicate utilizing open, yet precisely defined, interfaces, they can be conceptually easily:uninstalled—if the functionality in the current form is no longer needed, e.g., when instead of RRM realized per single BS, the O-RAN Service Provider (OSP) decides to apply an advanced coordinated solution;upgraded—when a new version of the module has been released, e.g., if the current vendor of the IM module equips it with new functionalities adjusted for a larger number of antennas installed on the mast;replaced—when an existing piece of software is removed and a new one delivered by other software vendors is installed;added—when the OSP decides to apply or test a new functionality, e.g., OSP wants to test new non-orthogonal multiple access (NOMA) schemes offered by some research entity.

All of the above-mentioned operations may be performed continuously and to some extent automatically. Thus, the Open RAN concept can modify the way the RAN software is delivered to and managed by the OSP—in the prospective free-market model, dedicated RAN functionalities (i.e., xApps) may be delivered by competing companies. Such an approach has the potential to reduce the price of RAN and improve the end-to-end quality as an effect of competitiveness, inherent to a free market.

Regarding AI modeling in O-RAN design, as per [17], ML training takes place in SMO or directly in non-RT RIC, while ML inference can be deployed in non-RT RIC or near-RT RIC where ML model updates and enrichment information are transferred from non-RT RIC to near-RT RIC via A1 interface. Examples of the AI/ML application within RIC include quality of experience (QoE) prediction, cell load prediction, prediction/detection of handover anomalies, and latency prediction. The example types of AI/ML algorithms for those applications include various versions of supervised learning, e.g., deep neural networks (DNN).

However, despite the evident advantages of Open RAN, various challenges have to be faced. First, this concept may be vulnerable to security threats, as the possibility of continuous and seamless updates of modules opens doors to various malicious activities. Next, the whole design process of RAN processing has to be redefined in such a way that Open RAN-compliant software will achieve similar or better results than the existing solutions, while allowing for software modularity and flexibility and openness of interfaces. Moreover, it is possible to apply AI tools to improve the management of installed applications. Let us consider the example of operation within near-RT RIC, where the conformance to standards and internal integrity has to be decided. xApp subscription management is required to identify and distribute the awareness of a new xApp being onboarded, and to allow data distribution among all xApps that subscribe to the data collection. Moreover, a conflict mitigation function is needed to resolve overlapping requests from xApps. For example, there may be a mobility load balancing (MLB) function working along with mobility robustness optimization (MRO). They both have a similar impact on user behavior. MLB may request the user to be moved from one cell to another due to high traffic load, while MRO at the same time may induce moving the user back as the handover boundary changes due to a high handover failure rate in the other cell. In such a case, the individual user will be subject to a “ping-pong” effect between the two cells. Thus, the conflict mitigation function’s job is to align the “to-be-invoked” actions to avoid such undesired behavior. This function judges what impact the potential action may have on the network if being executed in the E2-Node. One important note here is that an individual xApp directly controls a particular functionality (e.g., handover function) at the RAN nodes (O-CUs or O-DUs) and requests certain actions to be made in those.

Of course, these three aspects (security, efficiency, and advanced AI-based management) are not exhaustive, as many other challenges of Open RAN may be identified. In the following chapter, however, we intentionally concentrate on the modularity, flexibility, and efficiency of the delivered function of traffic steering.

## 4. Traffic Steering Use Case Analysis

The O-RAN Alliance specifies the use cases (UC) to be addressed by xApps, rApps and RIC 21, and defines the policies by which the algorithms designed to support the use cases can be controlled. The use cases are prioritized as per MNOs’ requirements. One such typical example is TS, the objective of which is to steer the user traffic through a specific cell, taking into account available schemes (such as handover, dual connectivity, carrier aggregation, license-assisted access, HetNet) and resources (such as multi-RAT, licensed and unlicensed carriers, etc.). Optimal traffic steering has been a challenge in wireless networks for many years, and numerous solutions have been proposed in the rich literature, addressing its various aspects [18,19,20,21]. As per [22], the challenge to be addressed by UC is that the typical TS schemes use radio conditions of a cell by treating all users in the same way and are limited to adjusting cell priorities and cell reselection/handover thresholds. The O-RAN Alliance aims at addressing the TS UC by customizing UE-centric strategies and proactive optimization by predicting network conditions and allowing MNOs to specify different objectives for traffic management depending on the scenario, and flexibly configure optimization policies. In this context, RIC is to control the adaptation of diverse scenarios and objectives and control TS strategies through AI/ML learning from user/network data.

Let us investigate a wireless network with open interfaces through which it is possible to implement network functionality externally. The goal of the conducted simulations and analyses is to show the possibility of applying the concept of Open RAN, where individual elements of RAN are replaced to optimize the operation of the entire network.

### 4.1. Simulation Setup

We consider a typical HetNet deployed, consisting of one high-power macro-BS and four small cells. To promote access to small cells, OSP can apply a dedicated power offset (achieving cell range extension). All cells may operate in two separated bands, i.e., a carrier frequency of 800 MHz with 5 MHz channel bandwidth, and 2 GHz with 10 MHz bandwidth. Only downlink (DL) transmission is considered. From the point of view of MNO, there is a cost *c* associated with the selection of lower and higher frequency bands, which reflect various kinds of loads for MNO, such as energy consumption cost, prices for a license, etc. Over the considered area, two types of users are randomly deployed with uniform distribution. A total of 80% are voice users (whose traffic is characterized by the constant and relatively low bit rate of ~250 kbps), and the rest are MBB users (with a constant and high bit rate of ~3 Mbps). The user is assumed to be in an outage when its achieved bit rate is below the required one. Let us also mention that our implementation is generic, i.e., we intentionally did not apply any of the existing platforms for RIC simulation. Our goal is to illustrate the benefits of xApp modularity supported by defined interfaces and the independence of the RIC platform.

### 4.2. xApp Implementation

The goal of this work was to make the TS functionality modular, changeable, controlled by an AI engine, and accessible by other external applications and by human administrators, as shown in Figure 3. We focused on the TS case, where OSP can apply various functioning policies, which are applied to independent xApps and realized in near-RT RIC. First, OSP may specify the rules of how the available two frequency bands should be utilized—this functionality is delivered to OSP in the form of xApp1. Second, OSP may define the preferences of usage of either the macro base station or the small cells (functionality delivered by xApp2). Finally, the priorities between the two types of users may be applied while offering them wireless services (defined within xApp3). Within each xApp, all necessary computations are performed to effectively apply the policies selected by OSP. 

In addition to that, non-RT RIC is equipped with an ML model to properly adjust policies to achieve certain goals based on non-RT measurements.

#### 4.2.1. xApp1 (Spectrum Management) for Frequency Band Selection

One of the prospective TS schemes is to offload traffic from a congested frequency band to a less occupied one. When moving the operating central frequency from higher bands to lower values, one may talk about the cell-zooming approach [23]. However, in a broader sense, various rules may be specified depending on the currently identified circumstances and needs of OSP. As we want to allow for RT modification of the policies, we define them in the form of specified tuples, which may be stored in the form of simple files, such as JSON or YAML, creating an application programming interface (API). Let us note that the policy can specify that, for example, the small cells should “prefer” larger bandwidths over costs or expected range, and the macro-cell should minimize its range. The word “prefer” should be treated loosely, allowing for various implementations by different applications. In the experiment, two policy options have been defined:CHEAP—where each cell should use the band with the lowest cost *c*;PERFORMANCE—where macro-cells should prefer a band with a higher range, and small cells should prefer a band with greater bandwidth.

#### 4.2.2. xApp2 (Cell Assignment) for User Assignment to BS

Analogously, one may specify the policies for user assignment to BS. The set of rules, in this case, specifies the “preference” of cell classes for different user classes. As an example, the policy can specify that small cells should “prefer” MBB users over voice users, and macro-cells should “prefer” voice users over MBB users. In this work, the following policies were considered:DEFAULT—where users are assigned to cells based on strongest received power;OFFLOADING—where MBB users should be preferred in small cells and voice users should be preferred in macro-cells;SEPARATING—where MBB users should be assigned only to small cells, and voice users should be assigned only to macro-cells.

#### 4.2.3. xApp3 (Resource Allocation) for Resource Scheduling

Finally, the last xApp is responsible for the definition of the prospective resource allocation strategy per BS. The policies in this case specify the “preference” of users of a specific class in terms of resource allocation. As an example, the policy can specify that MBB users will have more bandwidth allocated than voice users. Four policies have been identified:EQUAL—where all users have an equal amount of bandwidth allocated;PREFER_VOICE—where voice users have a larger bandwidth allocated (proportionally);PREFER_MBB—where MBB users have a larger bandwidth allocated (proportionally);RESERVE—where a specific user class has a reserved portion of bandwidth that can be used only by users of this class.

To be able to measure the performance of the system, the default setup of the network has been defined, where the cheapest frequency band is selected, the users are assigned to the cell based on received signal power, and the radio resources are allocated equally among the users.

### 4.3. System Training

Once the applications have been defined and implemented, we verified their functioning in all configurations. Thus, the performance of each xApp has been tested in terms of observed rate and outage probability, averaged over numerous user location deployments. Moreover, various joint configurations have been tested, e.g., simultaneous installation of xApp1 and xApp2. The achieved results have been presented and stored in the form of complementary cumulative distribution functions (CCDF). Selected results are also shown in Figure 3 for illustrative purposes only. By analyzing each particular plot, one can observe that while in some situations it is worth applying a certain policy, it is not that beneficial in other cases. Thus, to select the most promising policy, either the system administrator should analyze the curves and decide on the best strategy or let AI tools do it based on predefined criteria. In our tests, we have applied a simple ML tool (sitting at the non-RT RIC)—logistic regression, which allows for the selection of the policy that reduces outage in the system. Figure 3 presents the trained models, or more precisely, achieved CCDFs, which are available in non-RT RIC and influence the policy choice by near-RT RIC. Having such a system, we have performed experiments proving the benefits of xApp modularity, as discussed in the next section.

### 4.4. Simulation Results

In our experiments, we did not concentrate on the performance of an individual xApp, but rather we focused on the modularity and flexibility of the whole O-RAN application. Thus, we specified three experimentation scenarios, for which we showed selected performance metrics (mean bitrate for MBB and Voice users and outage probability) as a function of time for a random but fixed deployment of users. These scenarios have been considered to observe the benefits of temporal installation or de-installation of certain xApps, or at least modification of their operating policies (see Figure 4). Please notice that the reference setup to each scenario is the one where specific TS, spectrum management, and cell assignment algorithm is applied, and it is not subject to any changes.

Scenario 1: We start with the default, reference setup with no active xApp; at the 50 s time stamp, xApp1 is started, with the default CHEAP policy (analogous to the default setup), and thus no changes are observed; at 100 s, the policy changes to PERFORMANCE, leading to some mean bitrate increase (for both MBB and Voice users) but at the expense of some outage degradation; at 150 s, the system returns to the prior setup. As both the policy within xApp1, as well as the xApp1 itself can be modified (or even replaced, uninstalled, etc.), this scenario illustrates the benefits of xApp modularity and flexibility. OSP will have the opportunity to install or modify the selected features of its system even for a short time and observe the achieved results. Such performance measurements can also be analyzed by the AI engine to adjust the software to an instantaneous network change.

Scenario 2: Here, we extend the previous case to the situation where two separate xApps are installed, we observe interactions between them and react accordingly; thus, at 50 s, xApp3 starts preferring MBB users over voice users, thus the bitrate of MBB uses increases, and for Voice users, it decreases, leading at the same time to an improvement of outage probability; at 100 s, xApp1 chooses PERFORMANCE, which in this network state (i.e., location of users, their requests for resources) causes performance degradation; thus, the system selects again the CHEAP strategies at 150 s. This scheme shows the benefits of flexibility available to OSP—it can select the most suitable setup of the installed xApps and react immediately when any performance deterioration is observed. Please note that it is also possible to change or apply new policies within each xApp in order to adjust them to the current OSP needs. Such flexibility may not be easily available in a static, non-O-RAN scheme. This scheme also shows the prospective challenge that OSP has to face, i.e., maintaining the conformance of installed applications. Thus, the application of AI tools may be necessary.

Scenario 3: This is the most complicated scheme in our investigation. We start with all considered xApps running, and xApp1 works in the PERFORMANCE mode. Next, at 50 s, xApp starts preferring MBB users (resulting in MBB bitrate increase at no other cost), and at 100 s xApp2 selects the OFFLOADING policy (which is associated with some additional cost for OSP, some deterioration of Voice bitrate but significant gain in outage probability), whereas at 150 s, xApp1 returns to the CHEAP mode. One can observe that by testing these variants, at the end, OSP can find the best setup. Such an approach offers OSP the possibility to dynamically adjust the system setup to the changing network state. As the immediate installation and modification of selected features are not easily available in a black-boxed approach, the modularity and flexibility can then be treated as a good performance improvement opportunity.

## 5. Discussion

In this paper, we provided an overview of the O-RAN concept with particular emphasis on the RIC platform and RRM algorithm implementation and interaction in the form of xApps, along with ML-equipped modules. Such a programmable and highly modular RAN structure has a significant impact on future wireless network implementation. Using the O-RAN concept offers the following benefits:adding intelligence to network with external entities (such as xApps or RIC);controlling RAN behavior by declarative policies;combination of various applications to realize certain objectives/strategies;a hierarchical and modular approach to resource management;flexibility and modularity of RAN;defining applications/xApps per use case basis.

Looking into the interactions between the example xApps developed in the due time of this work, namely Cell Assignment, Spectrum Management, and Resource Allocation, the following holds:This approach increases the flexibility of controlling radio resources within the whole Traffic Steering use case, by playing with the radio resources on three levels;Combining the policies for the different xApps offers a possibility to optimize the usage of radio resources according to the selected strategy;Hierarchization allows for the separation of the concerns and focusing on a specific item;Modularization allows for expanding the framework with new xApps in a plug-and-play manner, which does not change the overall structure.

Summarizing the achieved results, one can observe that the possibility of installing/uninstalling the xApp jointly with the tailored method for selecting the best policy creates promising ways for OSP to improve the network performance depending on its (current) needs. The software modularity allows for fast adjustments of network functioning, thus leading to efficiency increase and adaptation to a particular situation or scenario.

In this paper, the authors provide a discussion on the case where all xApps operate for the purpose of a single use case. This, of course, can be different in real-life networks, where there are multiple use cases to be served at the same time, e.g., interference management together with massive MIMO and QoE optimization with network slicing resource assurance to multiple slices, resulting in having multiple different xApps running in parallel. A different situation could also arise when various xApps for the same functionality are used (e.g., Handover Optimization), but for different applications/scenarios (e.g., V2X, HetNet, Macro, UAV), thus resulting in different algorithms, which need to be selected properly for actual UEs. In such scenarios, where there are a multitude of various xApps running at the same time for the same network part, the role of the RIC itself becomes even more significant to avoid instability/oscillations in the network and properly select the actual xApp to control the individual UE or application, as an individual xApp controls the individual functionality at the E2-Node as exposed by this node.

Regarding the machine learning application within the Open RAN design, one example is the prediction of latency performed at RIC. Based on the predicted latency, near-RT RIC could, e.g., request O-DU to modify the scheduling priority of an individual user or primary/secondary cell reselection. One such application scenario is ultra-reliable and low latency communications (URLLC), where keeping low and/or predictive latency is one of the key requirements.

In addition to that, management of the network slices is one of the key applications of Open RAN, where dynamic setting up, modification, scaling and deleting of a network slice requires such an approach to virtualized and dynamically managed RAN [24].

## Figures and Tables

**Figure 1 sensors-21-08173-f001:**
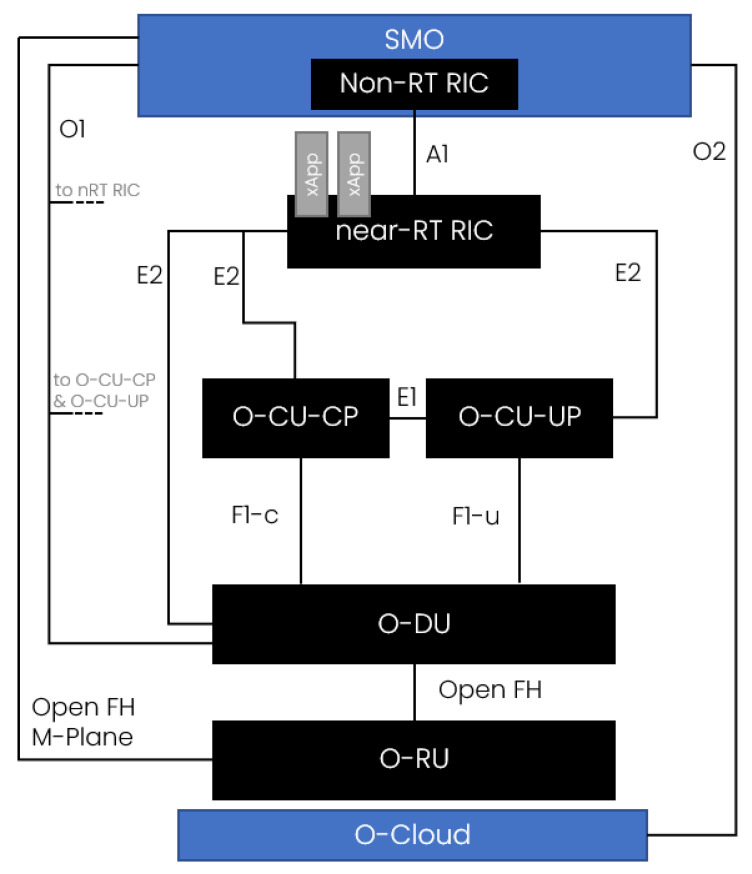
O-RAN Architecture.

**Figure 2 sensors-21-08173-f002:**
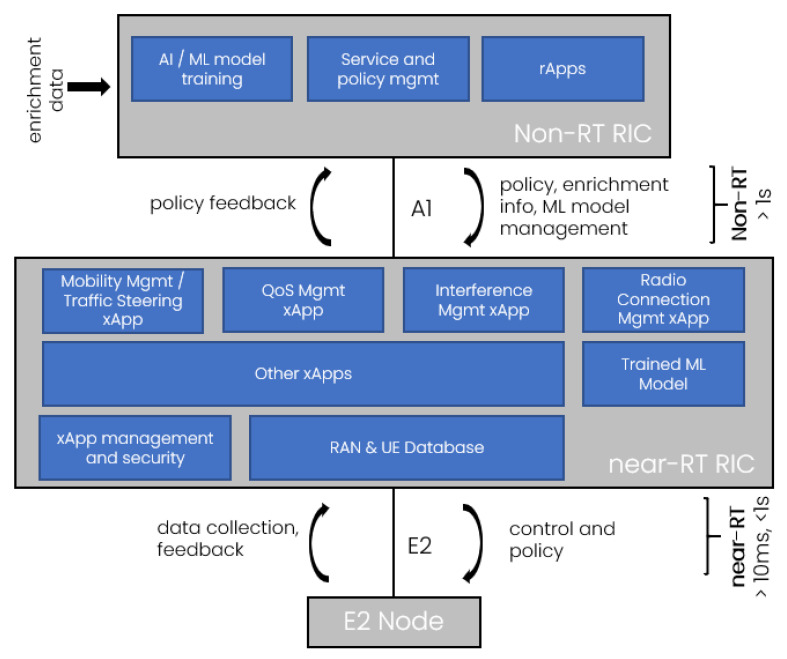
RAN Intelligent Controller: non-RT RIC and near-RT RIC.

**Figure 3 sensors-21-08173-f003:**
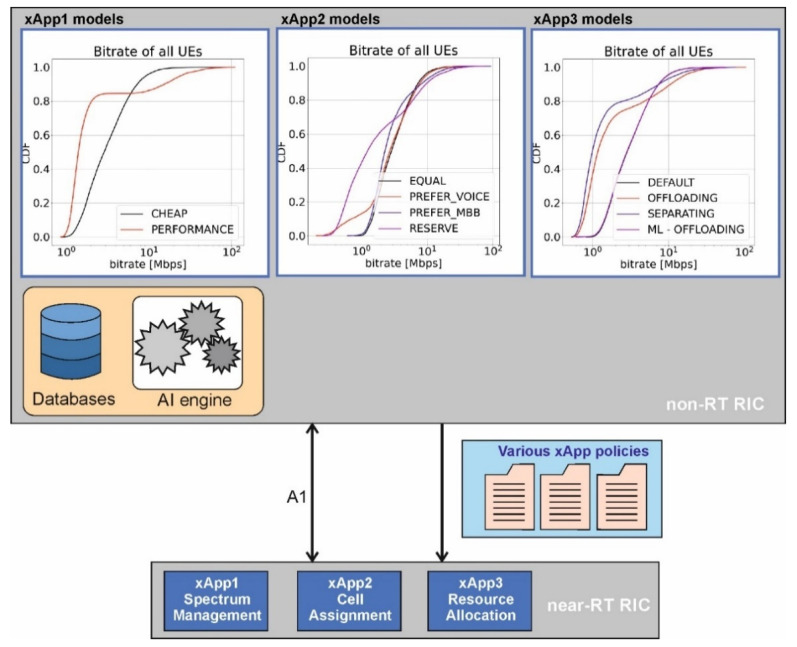
Mapping of xApps installations to RICs (xApp1—Spectrum Management, xApp2—Cell Assignment, xApp3—Resource Allocation).

**Figure 4 sensors-21-08173-f004:**
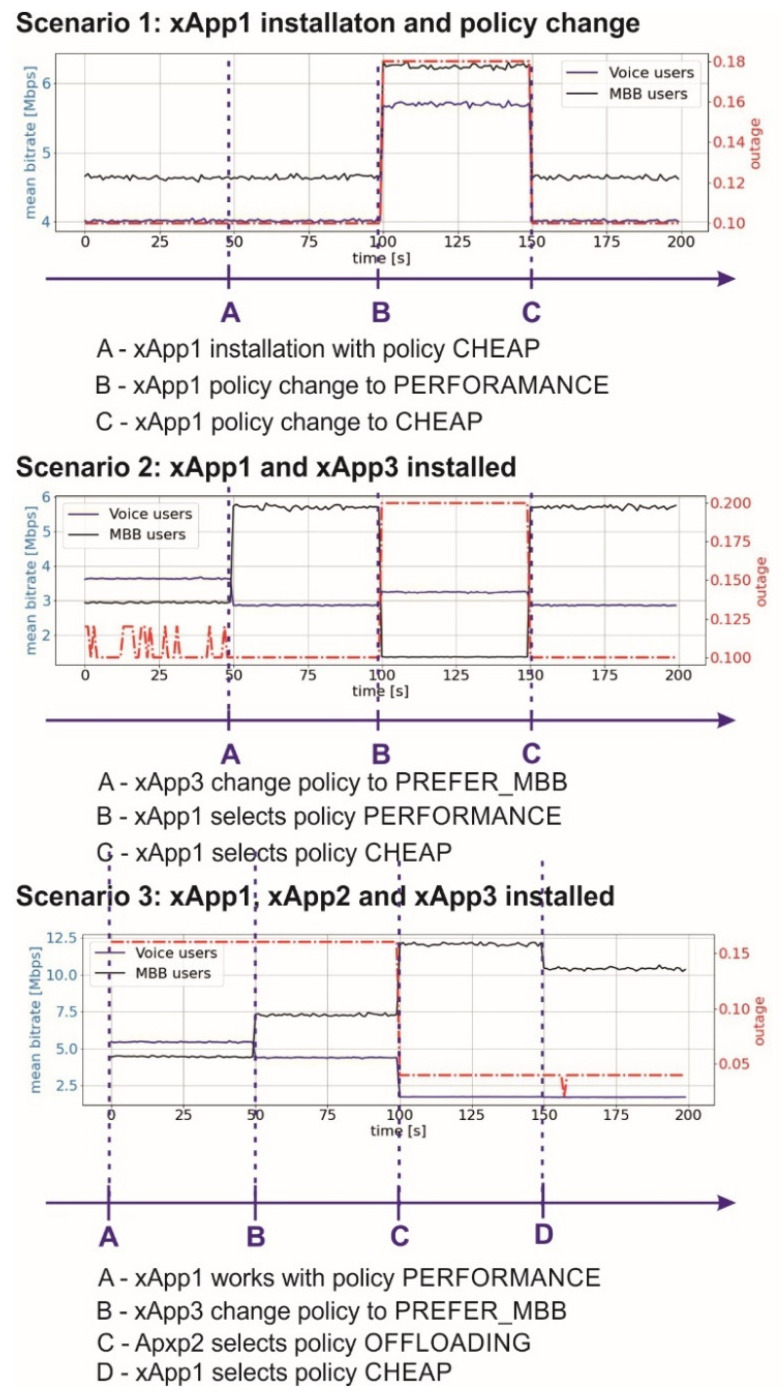
Achieved results for three considered experimentation schemes.

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
