# Peer review of "Toward Modular and Flexible Open RAN Implementations in 6G Networks: Traffic Steering Use Case and O-RAN xApps"

_sensors, 2021, doi:10.3390/s21248173_

Round 1
Reviewer 1 Report
This is a good overview paper that is overall well written. However, the manuscript is in need of improvements before it can be considered for publication:
- The work is not well enough embedded in the current literature, the number of references is not providing a bridge from 5G to 6G.
- There is no consideration for URLLC control loops as applications scenario, which will be one of the major future drivers past R16
- Some more details for training and simulations would be helpful (duration for training, stability, etc)
- The discussion is limited to bullet points, which is also attributed to the limited literature review. There should be more depth in the overall paper, rather than just showing the ORAN benefits (which are there!)
Author Response
Dear Editors and Reviewers,
Herewith, we submit the revised version of the paper entitled: "Toward Modular and Flexible Open RAN Implementations in 6G Networks: Traffic Steering Use Case and O-RAN xApps”. We put much effort to address all issues raised by the reviewers, and thus we hope that the current version is of better form. Please find detailed responses to the Reviewers’ comments and the description of improvements made in the revised paper. We attach the file with the changes marked in color.
With our best regards
Marcin Dryjański, Łukasz Kułacz, Adrian Kliks
Reviewer 1
Comment:
English language and style are fine/minor spell check required
Reply:
We have sent the paper to the English translator (human) to verify and improve the style
Comment:
This is a good overview paper that is overall well written. However, the manuscript is in need of improvements before it can be considered for publication.
Reply: Thank you for your positive opinion. Below one may find how we have addressed all the suggestions.
Comment
The work is not well enough embedded in the current literature, the number of references is not providing a bridge from 5G to 6G.
Reply: Thank you for this comment. Indeed the bridge was not there, thus we have added 5 new references and some more text in the intro, trying to better connect the 5G word with 6G visions.
Comment
There is no consideration for URLLC control loops as applications scenario, which will be one of the major future drivers past R16.
Reply: Thank you for this comment. Indeed, the importance of URLLC is very high now and will be high in the future. This, we have added a paragraph in the discussion section and also added some comments on network slicing.
Comment
Some more details for training and simulations would be helpful (duration for training, stability, etc)
Reply: Indeed, this part was missing. A few additional details about simulation and the training process are added.
Comment
The discussion is limited to bullet points, which is also attributed to the limited literature review. There should be more depth in the overall paper, rather than just showing the ORAN benefits (which are there!)"
Reply: Following this suggestion, the discussion section is expanded by 3 new paragraphs

Reviewer 2 Report
1, In Line33 on page1 and Line121 on page3, there should be “1,2” and “8,9”.
2, Some abbreviations need to be clarified, e.g., O-CU-CP and O-CU-UP in figure 1. The authors are suggested to provide an abbreviations table.
3, In section 3, the authors are suggested to provide some examples of applying AI tools in the RIC.
4, It is suggested to provide some previous researches about TS schemes.
5, In this paper, the authors consider all xApps about the TS problem, will it be different if some other objectives are considered simultaneously?
Author Response
Dear Editors and Reviewers,
Herewith, we submit the revised version of the paper entitled: " Toward Modular and Flexible Open RAN Implementations in 6G Networks: Traffic Steering Use Case and O-RAN xApps”. We put much effort to address all issues raised by the reviewers, and thus we hope that the current version is of better form. Please find detailed responses to the Reviewers’ comments and the description of improvements made in the revised paper. We also attach the file with changes marked in color.
With our best regards
Marcin Dryjański, Łukasz Kułacz, Adrian Kliks
Reviewer 2
Comments:
In Line33 on page1 and Line121 on page3, there should be “1,2” and “8,9”.
Reply: Thank you for pointing this issue, we have checked the paper for any possible editorial flaws.
Comments:
Some abbreviations need to be clarified, e.g., O-CU-CP and O-CU-UP in figure 1. The authors are suggested to provide an abbreviations table.
Reply: Indeed, we have added the glossary of acronyms at the end of the paper
Comment:
In section 3, the authors are suggested to provide some examples of applying AI tools in the RIC.
Reply: As the RIC is the entity responsible for controlling and managing, there is no limit in the tools that could be applied here. In that context, we have added a paragraph in chapter 3
Comment:
It is suggested to provide some previous researches about TS schemes.
Reply: Thank you for this comment, we have added some new example-references, showing the evolution of the traffic steering concept in various contexts.
Comment:
In this paper, the authors consider all xApps about the TS problem, will it be different if some other objectives are considered simultaneously?"
Reply: Indeed, this is a very important question. Definitely, the performance of the TS xApp will highly depend on the other objectives processed simultaneously. To address this point, we added a paragraph at the end of the conclusions.

Round 2
Reviewer 1 Report
The authors have incorporated the suggested changes into the revision. While I hoped for a more substantial inclusion of related works, the current state is publishable.